# Impact of Chromium Picolinate on Breast Muscle Metabolomics and Glucose and Lipid Metabolism-Related Genes in Broilers Under Heat Stress

**DOI:** 10.3390/ani15192897

**Published:** 2025-10-03

**Authors:** Guangju Wang, Xiumei Li, Miao Yu, Zhenwu Huang, Jinghai Feng, Minhong Zhang

**Affiliations:** 1State Key Laboratory of Animal Nutrition, Institute of Animal Science, Chinese Academy of Agricultural Sciences, Beijing 100193, China; guangju.wang@wur.nl (G.W.); llxiumei93@163.com (X.L.); 82101211223@caas.cn (M.Y.); 2019105026@njau.edu.cn (Z.H.); fengjinghai@caas.cn (J.F.); 2Adaptation Physiology Group, Wageningen University and Research, 6700 AA Wageningen, The Netherlands

**Keywords:** trace minerals, metabolic adaptation, muscle metabolite profiling, lipid homeostasis, feed additive strategy

## Abstract

**Simple Summary:**

This study examined the role of dietary chromium in broiler chickens under heat stress, with special attention to breast muscle metabolism. Chromium supplementation improved growth and reduced fat while lowering stress hormones and improving blood sugar balance. Metabolomic analysis of breast muscle showed that chromium changed key energy and amino acid metabolites, indicating better energy use and fat transport. These results suggest that chromium helps protect muscle metabolism and growth against heat stress.

**Abstract:**

The aim of the present study is to evaluate the impact of chromium (Cr) supplementation on glucose and lipid metabolism in breast muscle in broilers under heat stress. A total of 220 day-old broiler chicks were reared in cages. At 29 days old, 180 birds were randomly assigned to three treatments (0, 400, and 800 µg Cr/kg, as chromium picolinate) and transferred to climate chambers (31 ± 1 °C, 60 ± 7% humidity) for 14 days. Growth performance, carcass traits, serum biochemical indices, fasting glucose and insulin, homeostasis model assessment of insulin resistance (HOMA-IR), as well as muscle metabolomic profiles and gene expression related to energy and lipid metabolism were analyzed. The results showed that, compared with the heat stress group, the groups supplemented with 400 and 800 µg Cr/kg showed higher dry matter intake and average daily gain, breast muscle ratio, and lower feed conversion ratio and abdominal fat ratio; chickens supplemented with 400 and 800 µg Cr/kg showed significantly lower serum corticosterone (CORT), free fatty acids, and cholesterol levels compared with the heat stress (HS) group (*p* < 0.05). Fasting blood glucose and HOMA-IR were also significantly reduced, while fasting insulin was significantly increased in the Cr-supplemented groups (*p* < 0.05). Metabolomic analysis revealed that Cr supplementation regulated lipid and amino acid metabolism by altering key metabolites such as citric acid, L-glutamine, and L-proline, and modulating pathways including alanine, aspartate, and glutamate metabolism, and glycerophospholipid metabolism. Furthermore, Cr supplementation significantly upregulated the expression of Peroxisome Proliferator-Activated Receptor Gamma Coactivator 1 α (*PGC-1α*), ATP Binding Cassette Subfamily A Member 1 (*ABCA1*), Peroxisome Proliferator-Activated Receptor α (*PPARα*), and ATP Binding Cassette Subfamily G Member 1 (*ABCG1*) in both the hepatic and muscle tissue. This paper suggested that chromium supplementation may enhance energy metabolism and lipid transport like the findings of our study suggested.

## 1. Introduction

Heat stress is an increasingly harmful consequence caused by global warming, which has a wide range of detrimental effects on poultry production which includes reducing feed intake, growth rate, and feed conversion ratio, suboptimal carcass quality; excessive abdominal adiposity [1,2,3], while also exerting a detrimental effect on the immunity and organ function in broilers [4,5,6].

PPAR is a member of the nuclear receptor family involved in regulating lipid metabolism, lipoprotein metabolism, and glucose homeostasis [7]. One of the subtypes, *PPARα*, plays a crucial role in modulating gene expression associated with both peroxisomal and mitochondrial fatty acid oxidation. It also influences genes involved in fatty acid synthesis, transport, oxidation, and storage. By doing so, it contributes to the reduction in lipid accumulation and is intricately linked to the regulation of glucose and lipid metabolism as well as overall energy homeostasis [8]. The activation of *PPARα* can increase hepatic insulin activity and reduce body fat deposition [9].

Chromium is an essential trace mineral commonly used as a dietary supplement for both humans and animals. Studies indicate that heat stress triggers the release of chromium from tissues, increases its excretion through urine, and decreases its retention, leading to chromium deficiency in the body [10]. As a result, the requirement for chromium becomes more critical. There is substantial evidence that heat stress can increase circulating concentrations of corticosterone in broilers. There is evidence showing that corticosterone can reduce insulin sensitivity in broilers, resulting in insulin resistance, which is associated with many diseases such as type II diabetes, obesity, and fatty liver disease [11]. In this case, chromium is known to enhance insulin sensitivity in broilers [12]. Supplementing the diet of broilers with chromium picolinate under heat stress has been shown to enhance growth performance [13], increase carcass yield [14], decrease abdominal fat deposition [15], improve the weights of immune organs [16], increase insulin, and decrease serum glucose [14]. These findings suggest that chromium supplementation is capable of alleviating the negative effects of heat stress partly through improving insulin sensitivity and modulating stress responses in broilers.

In fast-growing broilers, the breast muscle (pectoralis major) is the most important skeletal muscle, accounting for 20–25% of live body weight, and serving as a highly metabolically active tissue [17]. Thus, its metabolomic analysis can provide detailed insights into glucose and lipid metabolism.

Our previous study showed that *PPARα* and *ABCA1* were reduced in expression in heat-stressed broilers, which suggested that the *PPARα* gene may play a critical role in the mechanism of lipid metabolism under heat stress [18]. However, in light of previous studies, it can be found that researchers mainly focus on the lipid metabolism of the chromium effect. However, it is largely unknown how Cr supplementation impacts the glucose metabolism and lipid metabolism in the skeletal muscle and these related genes. Additionally, it remains unclear whether the chromium supplementation can impact the subtype of *PPAR*, which in turn regulates glucose metabolism and lipid metabolism.

Therefore, the current study aimed to explore the effect of chromium on growth performance, blood parameters, breast muscle metabolism, and related gene expression under heat stress in broilers. We hypothesized that chromium dietary supplementation could affect glucose and lipid metabolism, and *PPARα* may play a role in mitigating the impact of chromium under heat stress.

## 2. Materials and Methods

### 2.1. Experimental Design, Animals and Housing

The experimental protocol had been approved by the Animal Experimental Welfare and Ethical Inspection Form of the Institute of Animal Science, Chinese Academy of Agricultural Sciences (permit number: IAS2020-112). A total of 220 day-old Arbor Acres male broiler chicks were reared in 22 one-tier elevated cages (0.8 m × 0.8 m × 0.4 m) to 28 days old. The corn-soybean meal starter diet that meets NRC (1994) requirements for Arbor Acres chickens for this period was obtained from Huadu Commercial Company (Beijing, China) and served ad libitum with through feeders (0.82 m long × 0.07 m wide × 0.06 m deep). The water was available with nipple drinkers. The set of all environmental parameters followed the Arbor Acres management handbook. At the age of 29 d, 180 healthy male birds with similar body weight (1150 ± 70 g) were selected and randomly assigned to three treatments: 0 (heat stress group), 400, and 800 µg Cr/kg of supplemental diet (chromium picolinate) and transferred to three climate chambers. The environmental chambers were of identical size, each containing 6 cages with 10 chickens each. A cage was seen as a replicate. All treatments were conducted under a constant ambient temperature of 31 ± 1 °C. The parameters of the chambers were set as follows: the humidity remained at 60 ± 7%, the rearing conditions corresponded to a temperature–humidity index (THI) of approximately 27.4 (ranging from 26.9 to 28.1), the airflow rate was under 0.5 m/s, and the ammonia concentration was controlled under 5 ppm. The lighting program was 24L:0D. The experiment lasted from 14 days to 42 days of age.

### 2.2. Diet

The basal diet was primarily composed of corn and soybean meal, and the composition is presented in Table 1. The heat stress group was subjected to the basal diet, while the diets of the other 2 groups were mixed with 400 and 800 µg Cr/kg chromium picolinate, respectively. Chickens were fed ad libitum and provided with free access to fresh drinking water throughout the experimental period. All diets were mixed thoroughly before feeding, and representative samples were analyzed for crude protein, metabolizable energy, and amino acid content to ensure diet consistency.

### 2.3. Measurements

The body weight was determined at the start and the end of the experiment. The dry matter intake was recorded on a daily basis during the experiment. The average daily gain (ADG), dry matter intake (DMI), and feed conversion ratio (FCR) were calculated. 2 birds per replicate were randomly selected for the body core temperature determination by a thermometer once a week during the experiment. At the end of the experiment, all chickens fasted for 12 h prior to the sampling. Thereafter, two chickens per replicate were selected based on their body weight closest to the cage average. The blood collection was performed using a blood collection needle in the wing vein. Subsequently, chickens were euthanized as Kulkari et al. described [15]. The left breast muscle and abdominal fat were weighed, and their respective proportions were calculated as a percentage of body weight. The liver was collected and placed in a cryotube, immediately transferred to liquid nitrogen, and subsequently stored in a −80 °C freezer.

### 2.4. LC-MS Analysis

Breast muscle samples were analyzed using hydrophilic interaction liquid chromatography (HILIC) with an ACQUITY UPLC BEH Amide column (2.1 × 100 mm, 1.7 μm, Waters, Milford, MA, USA). Electrospray ionization (ESI) was applied in both positive and negative modes. The mobile phase system consisted of 25 mM ammonium acetate and 25 mM ammonium hydroxide in water (A) and acetonitrile (B). The gradient elution began with 85% B for 1 min, gradually decreased to 65% over 11 min, quickly dropped to 40% within 0.1 min, and held for 4 min, followed by an increase to 85% within 0.1 min and a re-equilibration period of 5 min. For reversed-phase liquid chromatography (RPLC), an ACQUITY UPLC HSS T3 column (2.1 × 100 mm, 1.8 μm, Waters) was used. In positive mode, the mobile phase consisted of water and acetonitrile, both containing 0.1% formic acid, whereas in negative mode, 0.5 mM ammonium fluoride was added instead. The gradient profile was: 1% B for 1.5 min, increased to 99% over 11.5 min, maintained for 3.5 min, dropped to 1% within 0.1 min, and then equilibrated for 3.4 min. The column was operated at 25 °C with a flow rate of 0.3 mL/min, and 2 μL of each sample was injected for analysis.

### 2.5. Blood Parameter Analysis

Serum samples were obtained by centrifugation at 3000× *g* for 20 min at 4 °C. The fasting blood glucose was measured by using the Roche glucometer (Accu-Chek Performa, Roche Diagnostics, Indianapolis, IN, USA). The fasting insulin was determined by an ELISA kit that was obtained from Nanjing Jiancheng Institute of Bioengineering (H203-1-2 Insulin Test Kit, Nanjing Jiancheng, Nanjing, China). The Homeostasis model assessment of insulin resistance (HOMA-IR) was calculated by a formulation as follows: HOMA-IR = FBG (mmol/L) × FINS (mU/L)/22.5. The levels of free fatty acids, cholesterol, and CORT were measured using an automatic biochemical analyzer (Hitachi 7600, Hitachi Corporation, Tokyo, Japan). All the procedures were carried out according to the protocol of the manufacturer.

### 2.6. qPCR

Total RNA from breast muscle and liver samples was extracted using TRIzol reagent (Invitrogen, Carlsbad, CA, USA) following the manufacturer’s protocol. cDNA synthesis was performed using the HiScript II 1st Strand cDNA Synthesis Kit (R211) from Vazyme Biotech (Nanjing, China). The concentration and purity of RNA were measured using a Nanodrop 2000 spectrophotometer (Thermo Fisher Scientific, Waltham, MA, USA), and samples with A260/A280 ratios between 1.8 and 2.0 were used for cDNA synthesis. Real-time quantitative PCR was performed on a LightCycler 96 instrument (Roche, Basel, Switzerland) using SYBR Green PCR Master Mix (Applied Biosystems, Foster City, CA, USA). Each 20 μL reaction contained 10 μL SYBR Green Master Mix, 0.5 μM of each primer, and 2 μL of cDNA. The cycling conditions were as follows: initial denaturation at 95 °C for 10 min, followed by 40 cycles of 95 °C for 15 s, 60 °C for 30 s, and 72 °C for 30 s. Transcript levels of *ABCA1*, *ABCG1*, *PPARα*, and *PGC-1α* were quantified in both liver and breast muscle tissues. Gene-specific primers were designed based on GenBank sequences (see Table 2), and *β-actin* was employed as the reference gene for normalization. Relative gene expression was calculated using the 2^-ΔΔCt method.

### 2.7. Statistical Analysis

All data were analyzed using a one-way ANOVA based on a single-factor experimental design, followed by Duncan’s multiple range test for post hoc comparisons, implemented in SAS 9.2 software (SAS Institute Inc., Cary, NC, USA). Results were expressed as mean ± standard deviation (SD), with statistical significance defined at *p* < 0.05. For metabolomics, orthogonal partial least squares discriminant analysis (OPLS-DA) was employed to differentiate between groups, and metabolites with a variable importance in projection (VIP) score ≥ 1 and *p* < 0.05 (from independent sample *t*-tests) were considered significantly different. Identified differential metabolites between the HS and NT groups were annotated using KEGG IDs and analyzed through MetaboAnalyst. The pathways with *p* < 0.05 were deemed significantly affected.

## 3. Results

### 3.1. Growth Performance

All chickens remained free from diseases, and no mortality occurred during the experiment. As shown in Table 3, the ADFI and ADG in the 400 and 800 µg Cr/kg groups were significantly higher than those in the HS group (*p* < 0.05); the FCR was lower than that of the HS group. Additionally, the 400 and 800 µg Cr/kg groups both exhibited a significant increase in breast muscle rate and a significant decrease in abdominal fat rate compared to the HS group (*p* < 0.05). However, no significant differences in these parameters were observed between the 400 and 800 µg Cr/kg groups (*p* > 0.05).

### 3.2. Metabolomics in Breast Muscle

The scores (OPLS-DA) plot in Figure 1 shows a clear separation between the two groups, indicating distinct metabolic profiles. In the OPLS-DA model, the positive mode scores plot yielded R^2^X = 0.470, R^2^Y = 0.999, and Q^2^ = 0.779, while the negative mode scores plot resulted in R^2^X = 0.359, R^2^Y = 0.994, and Q^2^ = 0.854.

A total of 91 metabolites were identified as significantly different between the HS group and the group receiving 400 µg Cr/kg, as determined by OPLS-DA (VIP ≥ 1) combined with an independent sample *t*-test (*p* < 0.05). Detailed information on these differential metabolites is provided in Table 4. Relative to the HS group, 35 differential metabolites were upregulated and 56 differential metabolites were downregulated in the 400 µg Cr/kg group. The key differential metabolites involved in metabolic pathways include citric acid, L-aspartic acid, L-glutamine, L-proline, and (R)-(+)-2-Pyrrolidone-5-carboxylic acid, phosphatidylcholine (14:0/20:4[8Z,11Z,14Z,17Z]), 9,10,13-trihydroxy-octadecadienoic acid, phosphatidylcholine (22:6[4Z,7Z,10Z,13Z,16Z,19Z]), phosphatidylserine (18:0/20:0), and a series of lysophosphatidylcholine (LysoPC) species, including LysoPC (16:0), LysoPC (20:4[5Z,8Z,11Z,14Z]), LysoPC (18:1[9Z]), LysoPC (17:0), LysoPC (18:0), LysoPC (*P*-18:1[9Z]), LysoPC (20:1[11Z]), and LysoPC (P-18:0). Compared with the HS group, the concentrations of citric acid, L-aspartic acid, L-aspartic acid, L-glutamine, L-proline, (R)-(+)-2-pyrrolidone-5-carboxylic acid, PC(14:0/20:4[8Z,11Z,14Z,17Z]), 9,10,13-Trihydroxy-octadecadienoic acid were higher, and others were lower in the 400 µg Cr/kg group (*p* < 0.05).

As shown in Figure 2, pathway analysis revealed that supplementation with 400 µg Cr/kg had a significant impact on 9 metabolic pathways compared with the HS group (*p* < 0.05). These pathways include alanine, aspartate, and glutamate metabolism; D-glutamine and D-glutamate metabolism; arginine biosynthesis; linoleic acid metabolism; glycine, serine, and threonine metabolism; glyoxylate and dicarboxylate metabolism; aminoacyl-tRNA biosynthesis; ABC transporters; and glycerophospholipid metabolism.

### 3.3. Blood Parameters

Table 5 shows that the levels of CORT, free fatty acids, and cholesterol in serum in the 400 and 800 µg Cr/kg groups were significantly lower (*p* < 0.05), compared with the HS group (*p* < 0.05), whereas there was no significant difference in the 400 and 800 µg Cr/kg groups (*p* > 0.05). As shown in Figure 3, the fasting blood glucose and HOMA-IR in the 400 and 800 µg Cr/kg groups were significantly lower than those in the HS group (*p* < 0.05), and the fasting insulin in the 400 and 800 µg Cr/kg groups was significantly higher than that in the HS group (*p* < 0.05), whereas there was no significant difference in the 400 and 800 µg Cr/kg groups (*p* > 0.05).

### 3.4. Genes Associated with Glycolipid Homeostasis

As shown in Figure 4, the expression of *PGC-1α, PPARα, ABCA1*, and *ABCG1* in the liver in the 400 µg Cr/kg group was significantly higher than in the HS group (*p* < 0.05). As shown in Figure 5, the expression of *PGC-1α, PPARα, ABCA1*, and *ABCG1* mRNA in breast muscle in the 400 µg Cr/kg group was significantly higher than in the HS group (*p* < 0.05), whereas there was no significant difference in *ABCG1* in the breast muscle between the two groups (*p* > 0.05).

## 4. Discussion

The current study was performed under heat stress at 31 °C for 4 weeks. Our findings demonstrated that dietary Cr provision, regardless of the dosage, enhanced the growth performance and mitigated the glucose and lipid metabolism disorders.

Provision of Cr in broiler has been proven to mitigate the adverse impact on performance caused by heat stress [19]. Evidence has indicated that the use of Cr supplementation as a nutritional strategy to alleviate growth performance [20] via improved nutrient metabolism [14,21], immune responses [22], antioxidant function, and stress response [20,23] in broiler chickens exposed to heat stress. Our results demonstrated that dietary supplementation with 400 and 800 µg Cr/kg significantly improved the DMI and ADG of broilers while reducing the FCR. Moreover, Cr significantly increased the breast muscle yield and decreased the abdominal fat rate, suggesting a beneficial effect on carcass composition and meat quality. These findings were similar to studies conducted by Sahin et al. [24] and Sahin et al. [19], who found feed intake and body mass gain were enhanced after provision of 200, 400, 800, or 1200 μg Cr as Cr pic/kg diet. Sahin et al. [24] found that abdominal fat was decreased in chicks supplemented with 4 × 105 μg Cr as Cr pic/kg diet. Zha et al. [25] found FCR to be decreased in chicks supplemented with 500 μg Cr as Cr pic/kg diet or 500 μg Cr as Cr nano/kg diet and found that chicks supplemented with 500 μg Cr as Cr nano/kg diet had increased breast muscle mass. However, the results of the current study are not consistent with those of Amatya et al. [26] and Moeini et al. [27]. The conflicting results among these studies could be attributed, in part, to the differing chromium sources and administration dosages.

In our study, Cr supplementation significantly lowered fasting blood glucose (FBG) and the HOMA-IR while markedly increasing fasting insulin levels. These results indicate that Cr enhances insulin sensitivity and glucose homeostasis under HS conditions. The reduction in HOMA-IR suggests that Cr mitigates insulin resistance, which is commonly observed during heat stress due to glucocorticoid overproduction and metabolic stress [28]. Chromium is generally accepted to be the active component in glucose tolerance factor, which increases the sensitivity of tissue receptors to insulin, resulting in increased glucose uptake by cells [29,30]. The observed improvements in growth performance and carcass traits may be attributed to the ability of Cr to enhance insulin sensitivity and promote efficient nutrient utilization, particularly glucose and lipid metabolism. Previous studies have reported that Cr, particularly in the form of chromium picolinate, chromium propionate, and chromium chloride, enhances insulin receptor activity and facilitates glucose transport, thereby improving feed efficiency and lean tissue deposition in poultry under thermal stress conditions [14,31]. Furthermore, Cr supplementation has been shown to reduce lipid peroxidation and oxidative damage caused by HS, thereby supporting better muscle development and reducing fat accumulation [32].

Heat stress can stimulate the hypothalamus–pituitary–adrenal axis, causing an increase in the blood corticosterone concentration [33]. Studies have shown that excessive secretion of glucocorticoids, such as corticosterone, can trigger oxidative damage leading to muscle protein hydrolysis and significantly affect glucose metabolism, lipid metabolism, and muscle development [34]. In the present study, broilers supplemented with 400 and 800 µg Cr/kg exhibited significantly lower serum levels of corticosterone (CORT), free fatty acids (FFA), and cholesterol compared to the control group. It is reported that increasing chromium supplementation in broilers linearly reduces serum corticosterone concentrations [10]. The decreased FFA and cholesterol levels might be attributed to decreased glucocorticoid secretion, as demonstrated in a study by Sahin et al. [24].

In this study, supplementation with 400 µg Cr/kg under heat stress altered metabolic profiles in breast muscle metabolomics. Citric acid plays a critical role in the TCA cycle [35]. In addition, the L-aspartic acid and L-glutamine are involved in the malate-aspartate shuttle system, which promotes energy production [36]. On the other hand, L-glutamine and L-proline are vital components of the amino acid metabolism pathway and are involved in the antioxidant system [37], suggesting that Cr might reduce amino acid degradation by maintaining oxidative stability. PC is an important component of the cell membrane. One of the possible explanations is that the Cr might promote choletone and PC external to the cell by activating the ABC transport protein pathway. 9,10,13-Trihydroxy-octadecadienoic acid is the product of linoleic acid metabolism and has anti-inflammatory and antioxidant effects [38]. Therefore, the elevated levels of citric acid, L-aspartic acid, and L-glutamine suggest that the TCA cycle, amino acid metabolism, and linoleic acid metabolism might be improved by Cr supplementation. In the KEGG results, the enrichment of pathways confirmed our supposition. Dietary supplementation with 400 µg Cr/kg significantly influenced nine metabolic pathways, including alanine, aspartate, and glutamate metabolism; D-glutamine and D-glutamate metabolism; arginine biosynthesis; linoleic acid metabolism; glycine, serine, and threonine metabolism; glyoxylate and dicarboxylate metabolism; aminoacyl-tRNA biosynthesis; ABC transporter pathways; and glycerophospholipid metabolism. We hypothesize that Cr might enhance amino acid metabolism by promoting the biosynthesis and utilization of key amino acids such as glutamine, arginine, and glycine, which are essential for protein synthesis, nitrogen balance, and antioxidant defense under HS conditions. Improved glycine, serine, and threonine metabolism may enhance the production of glutathione, a critical antioxidant, thereby reducing oxidative stress and cellular damage caused by HS. Similarly, the upregulation of linoleic acid metabolism indicates enhanced lipid mobilization and utilization, which may contribute to the reduction in abdominal fat rate observed in Cr-supplemented broilers.

Our previous results indicated that heat stress can reduce *PPARα* level in the liver and suggested that *PPAR* might play an intermediate role in regulating lipid metabolism in broilers [18]. In the present study, the results indicated that the 400 µg Cr as pic/kg supplementation increased *PPARα* level to a large extent, which indicated that it helped to alleviate the adverse effects of heat stress on lipid metabolism. The beneficial effects of Cr on metabolic pathways are likely mediated through the activation of energy-regulating signaling cascades, such as the AMP-activated protein kinase (AMPK) and peroxisome proliferator-activated receptor gamma coactivator-1 alpha (*PGC-1α*) pathways. Previous studies have demonstrated that Cr can activate *AMPK*, a key cellular energy sensor, which in turn promotes glucose uptake, fatty acid oxidation, and mitochondrial biogenesis [39]. The increased expression of *PGC-1α, PPARα, ABCA1*, and *ABCG1* observed in the current study further supports this mechanism, indicating that Cr enhances hepatic and muscular energy metabolism and lipid transport. Moreover, the significant upregulation of *ABCA1* and *ABCG1* in the liver suggests that Cr enhances cholesterol efflux and reverse cholesterol transport (RCT). *ABCA1* and *ABCG1* are membrane transporters responsible for exporting excess cholesterol to high-density lipoproteins (HDL), facilitating its transport to the liver for clearance [40]. The increased expression of these transporters indicates that Cr promotes cholesterol homeostasis and reduces the risk of hepatic lipid overload under HS conditions. Notably, the elevated expression of *ABCA1* and *ABCG1* in the liver suggests that Cr promotes cholesterol efflux and lipid clearance [41], contributing to improved lipid homeostasis under heat stress, which is consistent with our earlier statement. Interestingly, the expression of *ABCG1* in the breast muscle was not significantly altered by Cr supplementation. This could be attributed to the low basal expression levels of *ABCG1* in skeletal muscle tissue or its relatively weaker role in lipid transport compared to the liver.

## 5. Conclusions

In conclusion, chromium supplementation might improve energy metabolism and lipid transport in the liver and muscles by upregulating the expression of genes related to glucose and lipid metabolism, thereby alleviating the damage to the glucose and lipid metabolic function and growth performance under heat stress conditions.

## Figures and Tables

**Figure 1 animals-15-02897-f001:**
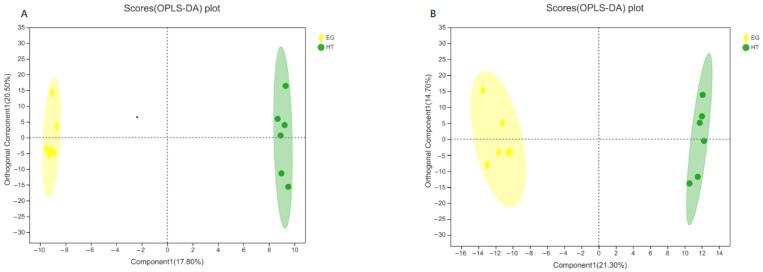
OPLS-DA score plots of metabolite profiles in broilers from the heat stress (HT) and 400 µg Cr/kg (EG) groups. (**A**) Positive ionization mode; (**B**) Negative ionization mode. Yellow represents the EG group, and green represents the HT group. Distinct clustering indicates metabolic differences between treatments.

**Figure 2 animals-15-02897-f002:**
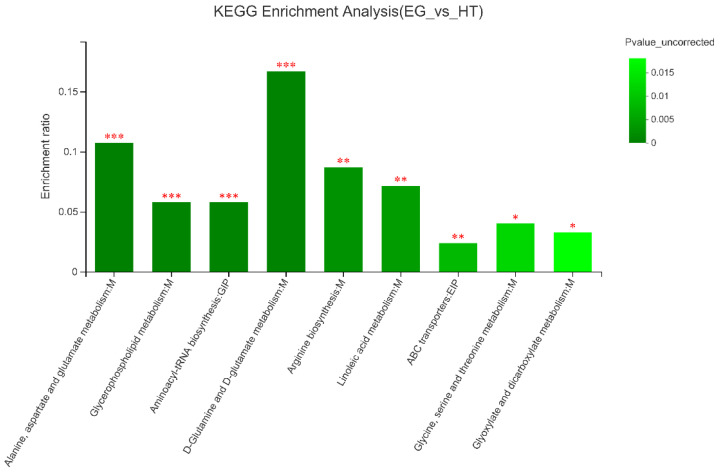
Pathway enrichment of the identified metabolites in broilers from the heat stress (HS) group and the 400 µg Cr/kg supplementation (EG) group. The x-axis means the pathway, while the y-axis represents the enrichment ratio, defined as the proportion of pathway-enriched metabolites to the total metabolites associated with that pathway. A higher enrichment rate signifies a greater degree of enrichment. The color gradient of the bars represents the significance of enrichment, and the dark color bars indicate more significant enrichment of the KEGG term. Statistical significance was determined based on *p*-values adjusted by FDR correction. Symbols denote the following: *p*-value < 0.001 is marked as ***, *p*-value < 0.01 is marked as **, and *p*-value < 0.05 is marked as *.

**Figure 3 animals-15-02897-f003:**
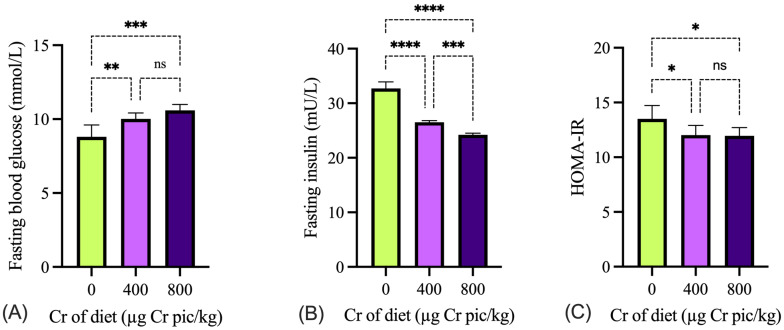
Effects of chromium picolinate supplementation on blood glucose (**A**), insulin (**B**) and HOMA-IR (**C**) in broilers under heat stress. Cr pic = chromium picolinate. Values are expressed as mean ± SD (n = 6). ns, not significant; * *p* < 0.05; ** *p* < 0.01; *** *p* < 0.001; **** *p* < 0.0001.

**Figure 4 animals-15-02897-f004:**
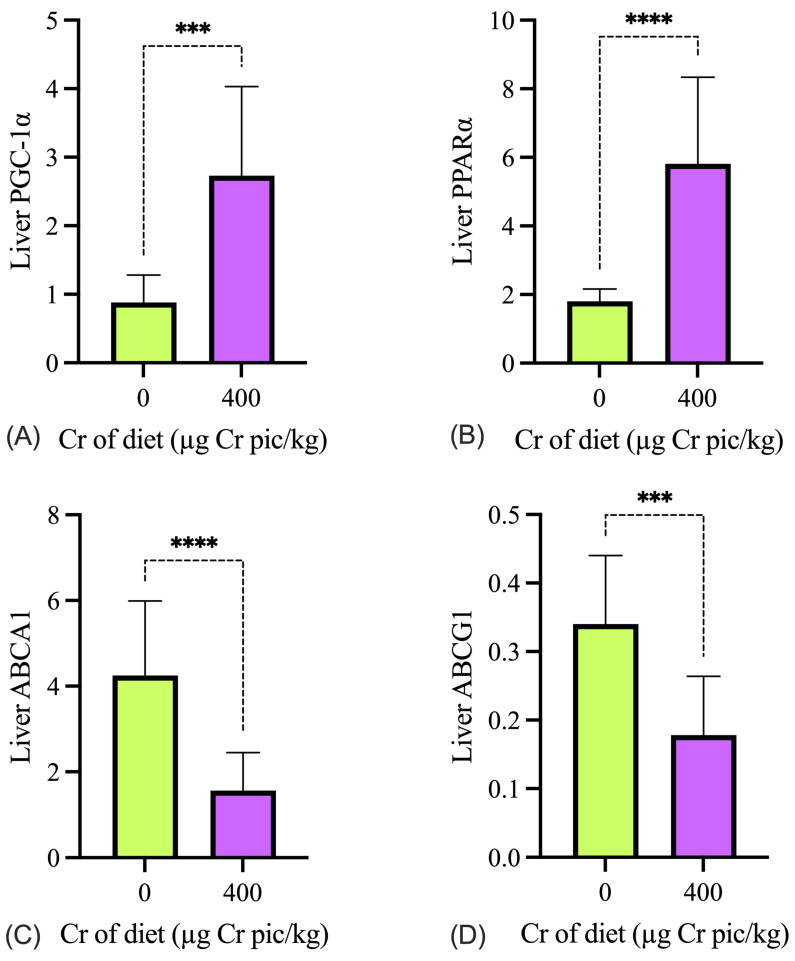
Effects of chromium supplementation on the mRNA expression of genes involved in glycolipid metabolism in the liver of heat-stressed broilers. The relative expression levels of *PGC-1α* (**A**), *PPARα* (**B**), *ABCA1* (**C**), and *ABCG1* (**D**) are shown. Cr pic = chromium picolinate. Values are expressed as mean ± SEM. *** *p* < 0.001; **** *p* < 0.0001.

**Figure 5 animals-15-02897-f005:**
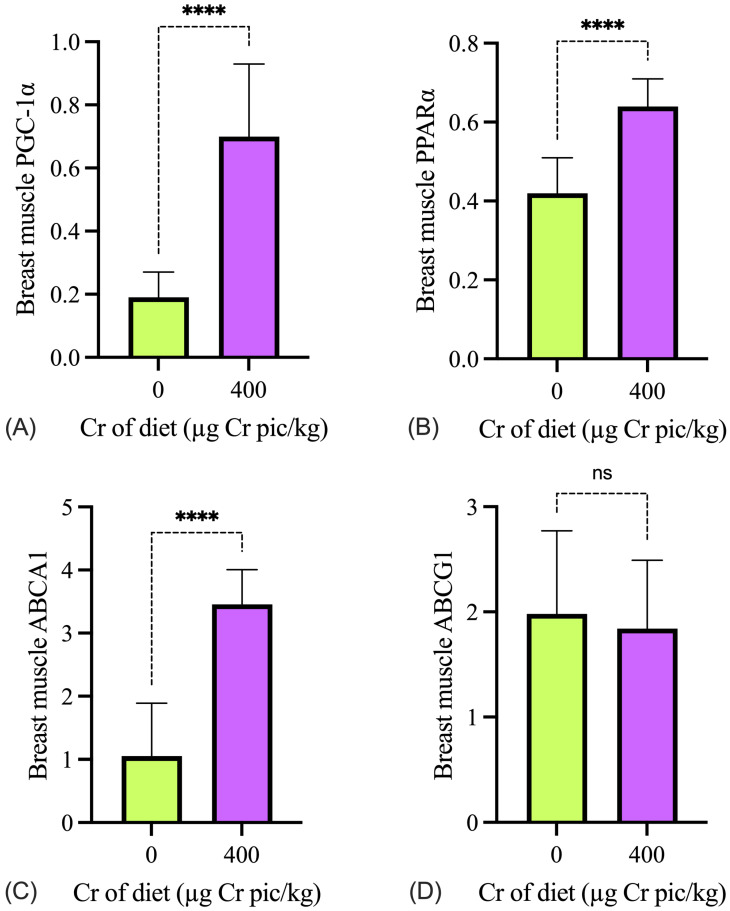
Effects of chromium supplementation on the mRNA expression of glycolipid metabolism genes in the breast muscle of heat-stressed broilers. The expression levels of *PGC-1α* (**A**), *PPARα* (**B**), *ABCA1* (**C**), and *ABCG1* (**D**) are shown. Cr pic = chromium picolinate.Values are expressed as mean ± SD (n = 6). ns, not significant; **** *p* < 0.0001.

**Table 1 animals-15-02897-t001:** Composition and nutrient levels of the basal diet (as-fed basis) %.

Items	Content
Ingredients	
Corn	56.51
Soybean meal	35.52
Limestone	1.00
Soybean oil	4.50
NaCl	0.30
dL-Methionine	0.11
CaHPO_4_	1.78
Premix ^1^	0.28
Total	100.00
Nutrient levels	
ME/(MJ/Kg)	12.73
Crude protein	20.07
Available Phosphorus	0.40
Calcium	0.90
Methionine	0.42
Lysine	1.00
Methionine + Cysteine	0.78

^1^ Premix provided the following per kg of the diet: VA 10,000 IU, VD3 3400 IU, VE 16 IU, VB1 2.0 mg, VK3 2.0 mg, VB2 6.4 mg, VB12 0.012 mg, VB6 2.0 mg, choline 500 mg, pantothenic acid calcium 10 mg, folic acid 1 mg, biotin 0.1 mg, nicotinic acid 26 mg, Zn (ZnSO_4_·7H_2_O) 40 mg, Mn (MnSO_4_·H_2_O) 80 mg, Fe (FeSO_4_·7H_2_O) 80 mg, I (KI) 0.35 mg, Cu (CuSO_4_·5H_2_O) 8 mg, Se (Na_2_SeO_3_) 0.15 mg.

**Table 2 animals-15-02897-t002:** Primers used for quantitative RT-PCR.

Primer Name ^1^	Primer Sequence 5′-3′ ^2^	Product Size (bp)	GenBank Accession Number
*β-actin*	F: CTGTGTTCCCATCTATCGT	270	NM_205518.2
	R: TCTTCTCTCTGTTGGCTTTG		
*ABCA1*	F: TCATCCACCGCCGCCACATT	223	NM204145
	R: GGCTGAGGAAGGCACTGAAGTC		
*ABCG1*	F: AACCAGTGGCTTGGATAGTGC	298	XM_025145525.1
	R: CCTTACCAGTCGGCTGTTCTG		
*PPARα*	F: CAAACCAACCATCCTGACGAT	22	NM_001001464.1
	R: GGAGGTCAGCCATTTTTTGGA		
*PGC-1α*	F: TGCAGCGCGATCTGAATG	110	NM_001006457.1
	R: GTTCTTGTCCTTGAGCCACTGAT		
	R: CAAGACTGACTGTGAAGGCATCCA		

^1^ β-actin, beta-actin. ^2^ F, forward; R, reverse.

**Table 3 animals-15-02897-t003:** Effects of chromium picolinate supplementation on growth performance of broilers under heat stress. ^1^

Items ^2^	0	400	800	*p*-Value
DMI/g	97.21 ^b^ ± 4.22	105.02 ^a^ ± 2.61	108.55 ^a^ ± 5.56	0.02
ADG/g	56.37 ^b^ ± 3.15	64.37 ^a^ ± 1.58	68.51 ^a^ ± 4.11	<0.01
FCRg/g	1.73 ^a^ ± 0.04	1.63 ^b^ ± 0.04	1.59 ^b^ ± 0.06	<0.01
Breast muscle rate (%)	17.85 ^b^ ± 1.27	19.18 ^a^ ± 1.18	18.90 ^a^ ± 0.7	0.01
abdominal fat rate (%)	0.77 ^a^ ± 0.20	0.59 ^b^ ± 0.22	0.56 ^b^ ± 0.10	0.02

^a,b^ Means within a column with different superscripts are different at *p* < 0.05. ^1^ All means are reported as means ± SD. ^2^ DMI = dry matter intake; ADG = average daily gain; FCR = DMI/ADG.

**Table 4 animals-15-02897-t004:** Differential metabolites involved in metabolic pathways in the metabolomics of breast muscle of HS and EG broilers.

Metabolites	VIP(HS&EG)	*p*-Value	Change
Citric acid	5.92	0.04	↑
L-aspartic acid	6.94	0.02	↑
L-glutamine	3.91	0.02	↑
L-proline	5.82	0.03	↑
(R)-(+)-2-Pyrrolidone-5-carboxylic acid	5.23	0.02	↑
Phosphatidylcholine (14:0/20:4[8Z,11Z,14Z,17Z])	3.94	0.05	↑
9,10,13-trihydroxy-octadecadienoic acid	4.69	0.002	↑
Phosphatidylcholine (22:6[4Z,7Z,10Z,13Z,16Z,19Z])	5.35	0.05	↓
Phosphatidylserine (18:0/20:0)	3.59	0.02	↓
LysoPC (16:0)	10.43	0.005	↓
LysoPC (20:4[5Z,8Z,11Z,14Z])	3.56	<0.001	↓
LysoPC (18:1[9Z])	5.97	0.03	↓
LysoPC (17:0)	8.95	0.002	↓
LysoPC (18:0)	6.93	0.02	↓
LysoPC (P-18:1[9Z])	8.38	<0.001	↓
LysoPC (20:1[11Z])	3.56	0.02	↓
LysoPC (P-18:0)	5.80	0.03	↓

**Table 5 animals-15-02897-t005:** Effects of 400 and 800 µg Cr/kg of chromium picolinate supplementation on blood parameters in broilers under heat stress.

Items	0	400	800	*p*-Value
Cholesterol (mmol/L)	4.80 ^a^ ± 0.11	3.57 ^b^ ± 0.52	3.21 ^b^ ± 0.26	<0.01
Free fatty acids (μmol/L)	106.68 ^a^ ± 3.24	89.06 ^b^ ± 1.46	106.68 ^b^ ± 3.24	<0.01
Corticosterone (ng/mL)	15.78 ^a^ ± 0.51	12.14 ^b^ ± 0.72	12.14 ^b^ ± 1.22	0.02

^a,b^ Means within a column with different superscripts are different at *p* < 0.05.

## Data Availability

The raw data supporting the conclusions of this article will be made available by the authors on request.

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
