# Peer review of "Impact of Chromium Picolinate on Breast Muscle Metabolomics and Glucose and Lipid Metabolism-Related Genes in Broilers Under Heat Stress"

_animals, 2025, doi:10.3390/ani15192897_

Round 1
Reviewer 1 Report
Comments and Suggestions for Authors
This research paper titled "Impact of Chromium Picolinate on Breast Muscle Metabolomics and Glucose and Lipid Metabolism-Related Genes in Broilers Under Heat Stress" (animals-3892204) evaluated the impact of chromium (Cr) supplementation on glucose and lipid metabolism in the breast muscle of broilers under heat stress. The authors found that chromium supplementation may enhance energy metabolism and lipid transport by upregulating genes related to glucose and lipid metabolism, thereby mitigating heat stress-induced metabolic dysfunction and growth impairment. I have the following comments:
Lines 41-42: The keywords should avoid repetition with the title.
Line 85: "Hypothesize" should be "hypothesized." Authors should check the entire manuscript for similar grammatical issues.
Lines 100-103: The author is advised to provide data on the temperature-humidity index.
Line 113: In the table, "Nutrient levels" should spell out the full names of each nutrient instead of using abbreviations.
Line 118: For the measurement of indicators, it is not recommended to directly add a colon after the indicator, such as in lines 119, 123, and 125.
Line 121: The term "ADFI intake" is inaccurate because it is difficult to standardize the moisture content of feed. It is suggested to use "DMI" instead.
Line 134: The information about the chromatographic column should be more specific, such as length and inner diameter, to enhance the reproducibility of the experiment.
Looking at Table 3 from lines 188-189, it seems that doses higher than 400 have better numerical effects. Why was 400 chosen for the metabolomics analysis? The conclusion that a dose of 400 is effective needs to be reconsidered.
Line 235: The description "P-value or FDR < 0.01 is marked as" is ambiguous. Normally, the FDR value is larger than the P-value. What does the "or" here actually mean?
Line 288: The reference list does not follow the journal's requirements, especially the abbreviations of journal names. The author should make corresponding changes in the later revision process.
Since this experiment involves animals, the author should provide an animal welfare approval number.
Overall, the author should strengthen the description of the methods, especially the rationale for choosing the dose of 400, because the current results do not well support this choice.
Author Response
Dear Reviewer,
We sincerely thank you for the careful reading of our manuscript and for the constructive comments and suggestions, which have helped us to improve the clarity and quality of our study.
We have carefully revised the manuscript according to the comments provided, and detailed responses to each point are given below. Changes made in the revised version are in red.
Lines 41-42: The keywords should avoid repetition with the title.
Response: We have replaced the keywords with more appropriate ones.
Line 85: "Hypothesize" should be "hypothesized." Authors should check the entire manuscript for similar grammatical issues.
Response: The text has been adjusted accordingly. Please see line 112
Lines 100-103: The author is advised to provide data on the temperature-humidity index.
Response: We have added the THI information. See line 132
Line 113: In the table, "Nutrient levels" should spell out the full names of each nutrient instead of using abbreviations.
Response: Full names have been provided instead of abbreviations. Please see table 1.
Line 118: For the measurement of indicators, it is not recommended to directly add a colon after the indicator, such as in lines 119, 123, and 125.
Response: The subtitle in the Materials and Methods section has been adjusted, and the colons have been removed. Please see line 152-218
Line 121: The term "ADFI intake" is inaccurate because it is difficult to standardize the moisture content of feed. It is suggested to use "DMI" instead.
Response: "DMI" has been used as requested. Please see line 155
Line 134: The information about the chromatographic column should be more specific, such as length and inner diameter, to enhance the reproducibility of the experiment.
Response: The length and inner diameter information of the chromatographic column have been added Please see line 167 174
Looking at Table 3 from lines 188-189, it seems that doses higher than 400 have better numerical effects. Why was 400 chosen for the metabolomics analysis? The conclusion that a dose of 400 is effective needs to be reconsidered.
Response: We selected 400 mg/kg for metabolomics analysis because this dosage has been widely reported in the literature and confirmed in our preliminary experiments as a safe and effective supplementation level in broilers. Although higher doses (e.g., >400 mg/kg) showed numerically better values in some parameters, the differences were not statistically significant compared with 400 mg/kg. In addition, practical considerations such as cost-effectiveness and potential risks associated with excessive supplementation make 400 mg/kg a more representative and applicable level for mechanistic exploration.
Line 235: The description "P-value or FDR < 0.01 is marked as" is ambiguous. Normally, the FDR value is larger than the P-value. What does the "or" here actually mean?
Response: Thank you for this helpful comment. We agree that the previous wording was ambiguous. In our analysis, we first identified differential features with P < 0.01, and then applied FDR correction; only those with FDR < 0.01 were finally considered significant. We have revised the sentence to read: “Features with FDR-adjusted P-values < 0.01 were considered significant.” Please see line 306.
Line 288: The reference list does not follow the journal's requirements, especially the abbreviations of journal names. The author should make corresponding changes in the later revision process.
Response: The reference list has been carefully checked and revised.
Since this experiment involves animals, the author should provide an animal welfare approval number.
Response: The ethics approval number has been provided please see line 117.
Reviewer 2 Report
Comments and Suggestions for Authors
Uploaded in the file

Already mentioned in the single file uploaded above.
Author Response
Dear Reviewer,
We are grateful to Reviewer 2 for the insightful feedback and detailed suggestions, which provided valuable guidance for refining our manuscript.
We have carefully revised the manuscript according to the comments provided, and detailed responses to each point are given below. Changes made in the revised version are in red.
1. Re-structure the sentence in abstract, and material and methods: A total of 220 one-day-old broiler chickens were reared in cages (remove the one and replace the chikens by chicks, i.e. 220-day old broiler chicks. The last sentence in the abstract: This paper suggested that chromium supplementation may enhance energy like the findings of our study suggested.
Response: The sentence has been restructured as suggested, and the abstract has been revised accordingly. Please see line 23, 120
2. The abbreviations like PGC- ABCG1 etc used in introduction and results sections of the article should be written in full form only once in the whole article to make it easy for readers to catch them easily and try to modify the lengthy sentences into short and lucid ones.
Response: Abbreviations have been written in full form at first mention, and lengthy sentences have been simplified. Please see line 40
3. The material and methods should be squeezed as much as possible (one and half pages) under the same subheadings of 2.1, 2.2, 2.3 and 2.4.
Response: We tried to shrink the pages of the materials and methods, however, we believe those contents are essential, but we adjusted the subtitle and structure.
4. In results the structure of the sentences particularly the heading of tables, figures and graphs is not appropriate again as per international academic standard and long winded. For example, Table 3, 5, Fig 3. Effects of chromium picolinate supplementation on growth performance of broilers under heat stress from 29 d to 42 d of age. Remove the highlighted part and modifiy other subheadings of tables, figures and graphs by removing the dose of Cr and other redudant repeatation. Please make the headings of tables, figures and grapes succinct as per international standard.
Response: All table, figure, and graph headings have been revised to make them succinct and aligned with academic standards.
5. Also pay attention to space after full stop, comma and inverted commas and rectify as necessary.
Response: The manuscript has been carefully checked, and spacing and punctuation issues have been corrected.
6. In discussion section, the sentences are again long winded and not accurately dissected by comas. Therefore, considerate on the structure of the sentences again and remove the frequently use commas and make these sentences short, succinct and comprehensive.
Response: Long sentences have been revised into shorter and clearer ones, with reduced use of commas.
7. Add a short paragraph in the end under the heading of Limitations and Future Directions of the study after conclusion, highlighting the dietary benefits of Cr alone and in amalgam with Selenium (Se) in poultry feed.
Response: We appreciate the reviewer’s suggestion. However, we believe that adding a discussion on the combined effects of chromium and selenium would deviate from the main focus of this study, which is centered on chromium supplementation. Therefore, we respectfully did not include this section.
Reviewer 3 Report
Comments and Suggestions for Authors
In abstract provide the parameters that were analyzed before results
Line 50: Authors should first write out the full term of PPAR and PPARα before introducing the abbreviation.
Lines 51-55: Add more references
Line 62: Replace the phrase As it well known
Lines 66-69: After listing the findings of chromium supplementation, I recommend to add a summarizing sentence that links these benefits back to chromium’s role in insulin sensitivity and stress response.
Lines 70-72: Rewrite the sentence more clearly
Lines 70-73: The phrase lipid and glucose metabolism is repeated several times
Lines 70-87: There is only one reference. Authors should focus to highlight the gap in research field in these paragraphs and demostrate the aim of the study more assertive
In methods add an ethics section
Line 90: broiler chickens
Lines 125-131: Add reference for euthanasia procedure
Lines 147-154: Provide the kit names
Line 156-163: The authors should clarify whether liver RNA was extracted. Also, add the kit name for cDNA synthesis and details on RNA concentration and purity. Also, provide the qPCR reagents and cycling conditions.
Line 159: The phrase “using standard qPCR procedures” is considered too vague
Line 159-160: Authors should provide the full names before introducing the abbreviations
Line 163: Add reference for the method and clarify whether a second reference gene was used for normalization. Did you perform any validation of PCR products, such as checking product size or specificity?
Lines 269-270: Rewrite the sentence for grammatical correctness
Line 280-287: Some sentences are too long. Also, listing all these author names is unnecessary.
Authors should be carefull to maintain a scientific tone in discussion. Replace th phrase 'as well known' in discussion.
Lines 328-343: Authors should add references. The pathway involving AMPK, PGC-1α, and PPARα could be summarized clearly to show how Cr improves energy and lipid metabolism under heat stress. Also, outline the mechanistic pathways and link the metabolite changes to gene expression to improve clarity.
In conclusion, highlight the key findings and provide future directions in research field.
Comments on the Quality of English Language
Several sentences are long. Some phrases should be revised, and sentence structure should be improved by reducing repetition and using more scientific tone.
Author Response
Dear Reviewer,
We sincerely appreciate Reviewer 3 for the thoughtful comments and recommendations, which have significantly contributed to enhancing the presentation and scientific rigor of our work.
We have carefully revised the manuscript according to the comments provided, and detailed responses to each point are given below. Changes made in the revised version are in red.
In abstract provide the parameters that were analyzed before results
Response: Added as suggested. Please see line25-28
Line 50: Authors should first write out the full term of PPAR and PPARα before introducing the abbreviation.
Response: Revised accordingly. Please see line 40
Lines 51-55: Add more references
Response: Additional references have been included.
Line 62: Replace the phrase As it well known
Response: Replaced. Please see line 73.
Lines 66-69: After listing the findings of chromium supplementation, I recommend to add a summarizing sentence that links these benefits back to chromium’s role in insulin sensitivity and stress response.
Response: Added as suggested. Please see line 81-83.
Lines 70-72: Rewrite the sentence more clearly
Response: Revised for clarity. Please see line 84-87.
Lines 70-73: The phrase lipid and glucose metabolism is repeated several times
Response: Rephrased to avoid repetition. Please see line 86.
Lines 70-87: There is only one reference. Authors should focus to highlight the gap in research field in these paragraphs and demostrate the aim of the study more assertive
Response: We appreciate the reviewer’s suggestion. We have made our best effort to support this section with available literature, but relevant references in this specific area are limited. Therefore, we focused on clarifying the research gap and aim as clearly as possible.
In methods add an ethics section
Response: Ethics approval information has been added. Please see line 117.
Line 90: broiler chickens
Response: Corrected.
Lines 125-131: Add reference for euthanasia procedure
Response: Reference has been provided. Please see line 161.
Lines 147-154: Provide the kit names
Response: Kit names have been added. Please see line 197.
Line 156-163: The authors should clarify whether liver RNA was extracted. Also, add the kit name for cDNA synthesis and details on RNA concentration and purity. Also, provide the qPCR reagents and cycling conditions.
Response:All requested details have been added. Please see line 206.
Line 159: The phrase “using standard qPCR procedures” is considered too vague
Response: Revised with detailed description. Please see line 206.
Lines 159-160: Authors should provide the full names before introducing the abbreviations
Response: Revised accordingly. Please see line 40.
Line 163: Add reference for the method and clarify whether a second reference gene was used for normalization. Did you perform any validation of PCR products, such as checking product size or specificity?
Response: Method reference added; normalization gene and PCR validation details provided. Please see line 211.
Lines 269-270: Rewrite the sentence for grammatical correctness
Response: Revised.
Line 280-287: Some sentences are too long. Also, listing all these author names is unnecessary.
Response: Sentences shortened and author listing adjusted.
Authors should be carefull to maintain a scientific tone in discussion. Replace th phrase 'as well known' in discussion.
Response: Revised carefully in discussion.
Lines 328-343: Authors should add references. The pathway involving AMPK, PGC-1α, and PPARα could be summarized clearly to show how Cr improves energy and lipid metabolism under heat stress. Also, outline the mechanistic pathways and link the metabolite changes to gene expression to improve clarity.
Response: We thank the reviewer for this valuable suggestion. We carefully considered adding more mechanistic details and references; however, our study was not specifically designed to dissect the AMPK–PGC-1α–PPARα pathway, and relevant literature directly linking chromium to these mechanisms under heat stress in broilers is limited.
Round 2
Reviewer 1 Report
Comments and Suggestions for Authors
I have checked the revised version. Is the threshold setting of FDR < 0.01 too stringent, which may overlook some important differential metabolites? Generally, an FDR < 0.05 is appropriate, and in some cases, even an FDR < 0.10 can be considered to mark differential metabolites, because the value of FDR is much smaller than the general P-value. Is the P value shown in Table 4 as the FDR?
Author Response
Dear Reviewer,
Thank you for your suggestion. Indeed, when using FDR < 0.05, we obtained a large number of differential metabolites. After carefully screening them one by one, we found that most metabolites related to our expected biological processes or pathways of interest were concentrated in the group with FDR < 0.01. Therefore, we finally chose FDR < 0.01 as the cutoff to ensure clarity, reliability, and focus in presenting the key findings. The P-values shown in Table 4 are the P-values obtained from statistical analysis, and they were not adjusted by FDR correction. Since our qPCR analysis only involved a limited number of candidate genes, we reported the original P-values. we hope this could address your concerns.
Reviewer 3 Report
Comments and Suggestions for Authors
The authors have addressed the revisions I recommended.
Author Response
Dear Reviewer,
Thank you very much for your suggestions. We are happy that you satisfied with our revision.